# A PRESCRIPTIVE THEORY FOR BRAIN-LIKE INFERENCE

## ABSTRACT

The Evidence Lower Bound (ELBO) is a widely used objective for training deep generative models, such as Variational Autoencoders (VAEs). In the neuroscience literature, an identical objective is known as the Free Energy Principle (FEP), hinting at a potential unified framework for brain function and machine learning. Despite its utility in interpreting generative models, including diffusion models, ELBO maximization is often seen as too broad to offer prescriptive guidance for specific architectures in neuroscience or machine learning. In this work, we show that maximizing ELBO under Poisson assumptions for general sequences leads to a spiking neural network that performs Bayesian posterior inference through its membrane potential dynamics. The resulting model, the iterative Poisson VAE ($i\mathcal{P}$-VAE), has a closer connection to biological neurons than previous brain-inspired predictive coding models based on Gaussian assumptions. Compared to amortized and iterative VAEs, $i\mathcal{P}$-VAE learns sparser representations and exhibits superior generalization to out-of-distribution samples. These findings suggest that optimizing ELBO, combined with Poisson assumptions, provides a solid foundation for developing prescriptive theories in NeuroAI.

## 1 INTRODUCTION

Optimizing the Evidence Lower Bound (ELBO) serves as a unifying objective for training deep generative models (Hinton et al., 1995; Dayan et al., 1995; Kingma & Welling, 2014; Rezende et al., 2014; Luo, 2022). Even when models don't explicitly reference ELBO, they're often optimizing objectives closely related to it (Luo, 2022; Kingma & Gao, 2023). This is directly paralleled by the Free Energy Principle (FEP) in neuroscience, which absorbs previous theoretical frameworks like Predictive Coding, Bayesian Brain, and Active Learning (Friston, 2005; 2009; 2010). FEP states that a single objective, the minimization of variational free energy, is all that is needed. Because this is equivalent to maximizing ELBO, it suggests a powerful unifying theoretical framework for neuroscience and machine learning (Friston, 2010).

However, in many ways, Free Energy (and by proxy, ELBO) is too general to be useful as a theory (Gershman, 2019; Andrews, 2021). In practice, the specific implementations of FEP predictive coding have been difficult to map directly onto neural circuits (Millidge et al., 2021a; 2022), struggling with negative rates and prediction signals that have not been observed empirically (Walsh et al., 2020; Millidge et al., 2022). Similarly, in machine learning, it is often discovered after the fact that a new objective is actually ELBO maximization (or KL minimization; Hobson (1969)) masquerading as something else (Kingma & Gao, 2023)—and not the other way around. If ELBO is "all you need," then why is ELBO not prescriptive?

One possibility, at least in neuroscience, is that ELBO's lack of prescriptive theory results from incorrect approximating distributions. In fact, most of the difficulty mapping predictive coding onto neural circuits has to do with terms that result from the Gaussian assumption (Millidge et al., 2022). In contrast, biological neurons are largely modeled as conditionally Poisson (Goris et al., 2014).

Recent work provides a potential prescriptive route: replacing Gaussians with Poisson distributions. To this end, Vafaii et al. (2024) introduced a reparameterization algorithm for training Poisson Variational Autoencoders ($\mathcal{P}$-VAE). They observed that replacing Gaussians in ELBO reduces to an amortized version of sparse coding, an influential model inspired by the brain that captures many features of the selectivity in early visual cortex (Olshausen & Field, 1996; 2004). $\mathcal{P}$-VAE learns sparse representations, avoids posterior collapse, and performs better on downstream classification

tasks. However, the authors identified a large amortization gap in $\mathcal{P}$-VAE (Vafaii et al., 2024), adding to a growing body of work that highlights limitations of amortized inference Cremer et al. (2018); Kim & Pavlovic (2021). A potential solution is to develop more general iterative inference solutions, or hybrid iterative-amortized ones (Marino et al., 2018; Kim et al., 2018).

Here, we extend the Poisson VAE to include iterative inference ("iterative $\mathcal{P}$-VAE," or i$\mathcal{P}$-VAE). This results in a generalization of predictive coding that maps well onto biological neurons. i$\mathcal{P}$-VAE implements Bayesian posterior inference via private membrane potential dynamics, resembling a spiking version of the Locally Competitive Algorithm (LCA) for sparse coding (Rozell et al., 2008). This solution avoids the major problems with predictive coding: there is no explicit prediction, neurons communicate through spikes, and feedback is modulatory—all consistent with real neurons (Gilbert & Li, 2013; Kandel et al., 2000). But how effective is i$\mathcal{P}$-VAE as a machine learning model?

We evaluate i$\mathcal{P}$-VAE in terms of convergence, reconstruction performance, efficiency, and out-of-distribution (OOD) generalization. We find that i$\mathcal{P}$-VAE converges to sparse posterior representations, outperforming other iterative VAEs (Kim et al., 2018; Marino et al., 2018).

**Contributions.** We introduce a new architecture, i$\mathcal{P}$-VAE, that accomplishes the following:

- Deriving the ELBO for sequences with Poisson-distributed latents results in a neural network that spikes, and performs predictive coding in the dynamics of the membrane potential.
- By reusing the same set of weights across iterations and utilizing sparse, integer spike counts, i$\mathcal{P}$-VAE is well-suited for hardware implementations and energy-efficient deployment.
- i$\mathcal{P}$-VAE demonstrates robust out-of-distribution generalization, excelling in both within-dataset perturbations and cross-dataset generalization.

Taken together, i$\mathcal{P}$-VAE is a powerful brain-inspired architecture that tightly maps onto biological neurons while outperforming much larger models in key objectives such as performance, parameter count, sparsity, and out-of-distribution generalization.

## 2 BACKGROUND AND RELATED WORK

**Generative models and ELBO.** Generative models learn to represent the data distribution, $p(\boldsymbol{x})$, typically by invoking latent variables $\boldsymbol{z}$, such that $p(\boldsymbol{x}) = \int p(\boldsymbol{x}|\boldsymbol{z})p(\boldsymbol{z})d\boldsymbol{z}$ (Bishop & Nasrabadi, 2006). The key challenge is computing, $p(\boldsymbol{z}|\boldsymbol{x})$, the posterior distribution of these latent variables given the data, which is typically intractable except for simple cases.

Variational inference offers a practical solution by introducing an approximate posterior $q_\phi(\boldsymbol{z}|\boldsymbol{x})$ parameterized by $\phi$ (Blei et al., 2017). The goal is to make this approximation as close as possible to the true posterior $p(\boldsymbol{z}|\boldsymbol{x})$. Ideally, one would minimize the KL divergence between $q_\phi(\boldsymbol{z}|\boldsymbol{x})$ and $p(\boldsymbol{z}|\boldsymbol{x})$, but since we cannot compute $p(\boldsymbol{z}|\boldsymbol{x})$ exactly, direct minimization is not feasible.

The Evidence Lower Bound (ELBO) provides a tractable objective that indirectly minimizes the KL divergence between the approximate and true posteriors. Specifically, the relationship is:

$$\log p(\boldsymbol{x}) = \underbrace{\mathbb{E}_{q_\phi(\boldsymbol{z}|\boldsymbol{x})}\left[\log \frac{p(\boldsymbol{x}, \boldsymbol{z})}{q_\phi(\boldsymbol{z}|\boldsymbol{x})}\right]}_{\text{ELBO}} + \mathcal{D}_{\text{KL}}\Big(q_\phi(\boldsymbol{z}|\boldsymbol{x}) \,\|\, p(\boldsymbol{z}|\boldsymbol{x})\Big) \tag{1}$$

Since $\log p(\boldsymbol{x})$ does not depend on $\phi$ and the KL divergence is non-negative, maximizing the ELBO effectively minimizes the intractable KL divergence (Hinton et al., 1995; Kingma & Welling, 2014; Rezende et al., 2014). Interestingly, even when generative models seem to optimize a different loss function, like diffusion models (Chan, 2024; Ho et al., 2020), they are often still performing KL minimization through the ELBO (Kingma & Gao, 2023; Luo, 2022).

**ELBO in Neuroscience.** The Evidence Lower Bound (ELBO) has an identical formulation in neuroscience, where it is referred to as the Free Energy (Friston, 2005; 2009; 2010). The Free Energy Principle (FEP) extends the framework of perception as inference (Alhazen, 1011–1021 AD;

Von Helmholtz, 1867; Mumford, 1992), drawing concepts from predictive coding (PC; Srinivasan et al. (1982); Rao & Ballard (1999)). Extensive research has explored how PC might be implemented by neurons (Boerlin et al., 2013; Millidge et al., 2021a), and PC has been applied in machine learning for predictive models (Lotter et al., 2017; Wen et al., 2018; Millidge et al., 2024).

Despite their neural inspiration, FEP is challenging to map directly onto neuronal circuits (Kogo & Trengove, 2015; Aitchison & Lengyel, 2017; Millidge et al., 2022). This difficulty results from assuming Gaussian for the approximate posterior and prior (Millidge et al., 2022). The Gaussian assumption results in models with explicit predictions or prediction errors, which have not been observed empirically (Mikulasch et al., 2023). Solutions also struggle with how to avoid negative firing rates due to subtraction operations (Bastos et al., 2012; Keller & Mrsic-Flogel, 2018). While leaky integrate-and-fire (LIF) circuits can be engineered to perform predictive coding (Boerlin et al., 2013), these implementations do not naturally arise from ELBO maximization, making the theory more postdictive than prescriptive. The related framework of sparse coding can be thought of as a form of predictive coding with a sparse prior Olshausen & Field (1996; 2004). A biologically plausible implementation of sparse coding, known as the locally competitive algorithm (LCA; Rozell et al. (2008)), results naturally in a dynamic update rule that resembles neural circuits. However, LCA relies on maximum a posteriori inference, which is restrictive if we aim to sample from the full posterior distribution.

**Bayesian posterior inference: iterative versus amortized.** In contrast to predictive coding, Variational Autoencoders (VAEs) introduced a computationally-efficient solution to maximize ELBO through *amortized* inference (Kingma & Welling, 2014; Rezende et al., 2014). Amortized inference uses a parameterized neural network (the "encoder" or "recognition" network) to produce the parameters of an approximate posterior, $q_\phi(z|x)$, in one shot. The term "amortized" reflects that the computational cost of inference is paid during training, not at test time, similar to cost distribution in accounting (Gershman & Goodman, 2014). While amortized inference is considered efficient, it can suffer from an *amortization gap*—the discrepancy between the approximate posterior provided by the encoder and the optimal variational parameters—which can be significant (Cremer et al., 2018).

To address the amortization gap, hybrid approaches have been developed that introduce iterative elements into the VAE framework (Marino et al., 2018; Kim et al., 2018; Marino et al., 2021). For example, Marino et al. (2018) proposed a method where the encoder network takes as input both the data sample $x$ and the gradients of the loss with respect to the variational parameters $\nabla_\lambda \mathcal{L}$, with $\lambda = \{\mu, \sigma^2\}$. Alternatively, semi-amortized inference (Kim et al., 2018) starts with an amortized initial estimate and refines it using stochastic variational inference (SVI) updates (Hoffman et al., 2013). Our method is closely related to these approaches, and we compare to them in the results.

Although VAEs and predictive coding are related through their optimization of ELBO (Marino, 2022), recent work has made that connection more explicit, demonstrating that classical predictive coding networks can be seen as a subclass of iterative inference in VAEs (Boutin et al., 2020). A key difference between our work and Boutin et al. (2020) is that they show the Rao & Ballard (1999) loss function arises from assuming a delta-function posterior in the ELBO. In our work, predictive coding naturally emerges in the dynamics of the log spike rates, which comes from a fairly general assumption of Poisson distributions.

**Poisson VAE.** A large body of literature in neuroscience has demonstrated that neuron spike counts are well described by a Poisson process over short counting windows (Goris et al., 2014). Building on this, Vafaii et al. (2024) introduced the Poisson Variational Autoencoder (P-VAE), which performs posterior inference using discrete spike counts. They developed a Poisson reparameterization trick and derived the ELBO for Poisson-distributed VAEs ($\mathcal{P}$-VAE).

In $\mathcal{P}$-VAE, the KL term penalizes firing rates, similar to sparse coding, and the ELBO, when paired with a linear generative model, reduces to amortized sparse coding. When trained on natural image patches, $\mathcal{P}$-VAE learns sparse solutions with Gabor-like basis vectors and latent sparsity, similar to sparse coding. While $\mathcal{P}$-VAE outperformed Gaussian VAEs in sparsity and downstream classification, the authors noted a significant performance gap with traditional sparse coding, likely arising from an amortization gap due to the lack of iterative updates. Our work builds upon $\mathcal{P}$-VAE, suggesting that Poisson is the right choice for parameterizing the distributions in ELBO (see Appendix B for a discussion).

## 3 INTRODUCING THE ITERATIVE POISSON VAE (IP-VAE)

In this section, we derive the ELBO for sequences with Poisson distributions. We show the resulting architecture (i$\mathcal{P}$-VAE) implements iterative Bayesian posterior inference with dynamics on the log rates. We relate this directly to membrane potential dynamics in a spiking neural network and show that it solves many of the implementation limitations of classic predictive coding.

**General setup.** We conceptualize iterative inference by starting with the more general framework of inference over a sequence (Chung et al., 2015). From there, we can treat iterative inference for images as a sequences of the same image repeated at all time points. This approach is appealing because dynamics emerge necessarily, and it builds a foundation for future work on dynamic sequences.

Consider a sequence of $T + 1$ observed data points, $\vec{\boldsymbol{x}} = \{\boldsymbol{x}_t : t = 0, \ldots, T\}$ where $\boldsymbol{x}_t \in \mathbb{R}^M$, and corresponding latent variables, $\vec{\boldsymbol{z}} = \{\boldsymbol{z}_t : t = 0, \ldots, T\}$, where each $\boldsymbol{z}_t$ is $K$-variate. We denote the full probabilistic generative model as the joint distribution, $p(\vec{\boldsymbol{x}}, \vec{\boldsymbol{z}})$. A reasonable starting assumption for modeling the physical world is Markovian dependence between consecutive data points (Van Kampen, 1992), resulting in the marginal distribution:

$$p(\vec{\boldsymbol{x}}) = \int p(\vec{\boldsymbol{x}}, \vec{\boldsymbol{z}}) \, d\vec{\boldsymbol{z}} = p(\boldsymbol{x}_0) \prod_{t=1}^{T} p(\boldsymbol{x}_t | \boldsymbol{x}_{t-1}), \tag{2}$$

where $p(\boldsymbol{x}_0) = \int p(\boldsymbol{x}_0 | \boldsymbol{z}_0) p(\boldsymbol{z}_0) d\boldsymbol{z}_0$, and $p(\boldsymbol{x}_t | \boldsymbol{x}_{t-1}) = \int p(\boldsymbol{x}_t | \boldsymbol{z}_t) p(\boldsymbol{z}_t | \boldsymbol{x}_{t-1}) d\boldsymbol{z}_t$. For our sequence data, the ELBO can be written as follows:

$$\begin{aligned} \log p_\theta(\vec{\boldsymbol{x}}) &\geq \mathbb{E}_{q_\phi(\vec{\boldsymbol{z}} | \vec{\boldsymbol{x}})} \left[ \log \frac{p_\theta(\vec{\boldsymbol{x}}, \vec{\boldsymbol{z}})}{q_\phi(\vec{\boldsymbol{z}} | \vec{\boldsymbol{x}})} \right] \\ &= \mathbb{E}_{q_\phi(\vec{\boldsymbol{z}} | \vec{\boldsymbol{x}})} \left[ \log p_\theta(\vec{\boldsymbol{x}} | \vec{\boldsymbol{z}}) \right] - \mathcal{D}_{\text{KL}} \left( q_\phi(\vec{\boldsymbol{z}} | \vec{\boldsymbol{x}}) \, \| \, p_\theta(\vec{\boldsymbol{z}}) \right) \\ &= \mathcal{L}_{\text{ELBO}}(\vec{\boldsymbol{x}}; \theta, \phi), \end{aligned} \tag{3}$$

where $p_\theta(\vec{\boldsymbol{z}})$ is a prior (either learned or fixed) over latents, and $p_\theta(\vec{\boldsymbol{x}} | \vec{\boldsymbol{z}})$ is the conditional likelihood distribution, which is computed via a decoder network. Model parameters, $(\phi, \theta)$—corresponding to the encoder and decoder networks of a VAE, respectively—are jointly optimized. Below we will express the ELBO for sequences when using the Poisson Variational Autoencoder framework.

**Iterative Poisson VAE.** To extend the $\mathcal{P}$-VAE to sequences, i$\mathcal{P}$-VAE needs to make explicit how the prior and posterior distributions update with each sample. The simplest starting point is assuming stationarity, implying that the posterior over the previous stimulus should act as a prior for the current one (although future extensions could extend to nonstationary signals such as videos with a more sophisticated update rule). Because of the Markovian assumption, the prior, $p(\vec{\boldsymbol{z}})$, then factorizes into the initial prior, $p(\boldsymbol{z}_0)$ and a product over all time steps:

$$p(\vec{\boldsymbol{z}}) = p(\boldsymbol{z}_0) \prod_{t=1}^{T} p(\boldsymbol{z}_t | \boldsymbol{x}_{t-1}) \tag{4}$$

The initial prior, $p(\boldsymbol{z}_0) = \mathcal{P}\text{ois}(\boldsymbol{z}_0; \boldsymbol{r}_0)$, is Poisson with learned prior rates, $\boldsymbol{r}_0 \in \mathbb{R}^K_{>0}$. Subsequent time steps have prior rates that depend on the stimulus from the previous time step, $p(\boldsymbol{z}_t | \boldsymbol{x}_{t-1}) = \mathcal{P}\text{ois}(\boldsymbol{z}_t; \boldsymbol{r}_t(\boldsymbol{x}_{t-1}))$. The approximate posterior factorizes as well:

$$q(\vec{\boldsymbol{z}} | \vec{\boldsymbol{x}}) = q(\boldsymbol{z}_0 | \boldsymbol{x}_0) \prod_{t=1}^{T} q(\boldsymbol{z}_t | \boldsymbol{x}_t, \boldsymbol{x}_{t-1}), \tag{5}$$

with initial posterior, $q(\boldsymbol{z}_0 | \boldsymbol{x}_0) = \mathcal{P}\text{ois}(\boldsymbol{z}_0; \boldsymbol{r}_0 \odot \boldsymbol{\delta r}(\boldsymbol{x}_0))$, and time-dependent posterior, $q(\boldsymbol{z}_t | \boldsymbol{x}_t, \boldsymbol{x}_{t-1}) = \mathcal{P}\text{ois}(\boldsymbol{z}_t; \boldsymbol{r}_t(\boldsymbol{x}_{t-1}) \odot \boldsymbol{\delta r}(\boldsymbol{x}_t))$, both parameterized as Poisson distributions. We follow the formulation in Vafaii et al. (2024), and define the posterior rates via an element-wise multiplicative interaction between $\boldsymbol{r}$ and some gain modulator, $\boldsymbol{\delta r} \in \mathbb{R}^K_{>0}$. This is a natural choice because rates must be positive, and without loss of generality, the relationship between two positive variables can be written in terms of a base rate, and a multiplicative gain on that base rate.

The conditional log-likelihood for i$\mathcal{P}$-VAE factorizes into a sum over individual sample likelihoods $\log p(\vec{x}|\vec{z}) = \sum_{t=0}^{T} \log p(x_t|z_t)$. The KL-term of the ELBO (eq. (3)) also factorizes:

$$\mathcal{D}_{\mathrm{KL}}\Big(q(\vec{z}|\vec{x}) \,\big\|\, p(\vec{z})\Big) = \mathcal{D}_{\mathrm{KL}}\Big(q(z_0|x_0) \,\big\|\, p(z_0)\Big) + \sum_{t=1}^{T} \mathcal{D}_{\mathrm{KL}}\Big(q(z_t|x_t, x_{t-1}) \,\big\|\, p(z_t|x_{t-1})\Big)$$

$$= r_0 \cdot f(\delta r(x_0)) + \sum_{t=1}^{T} r_t(x_{t-1}) \cdot f(\delta r(x_t)),$$
(6)

where $\cdot$ represents a vector dot product, and $f(y) = 1 - y + y\log y$ is applied element-wise. Because rates are positive, the KL term penalizes large rates, acting like a sparsity penalty (Vafaii et al., 2024). The remaining sections describe how we specify the multiplicative gain, $\delta r$, which results in adaptive Bayesian posterior updating in the dynamics of the model.

**Bayesian posterior updates using membrane potential dynamics**   Because rates are positive and prior and posterior rates interact multiplicatively, it is difficult to implement dynamic updates directly on rates. A natural solution is to define updates on log rates, $u(t) := \log r(t)$, with $\mathbb{R}^K$ as our state space for a $K$-dimensional latent space.

Dynamic updates on log-rates is both a mathematical convenience and biologically realistic. Because of internal noise, the spike threshold of real neurons is best modeled as an expansive nonlinearity like an exponential (Priebe et al., 2004; Fourcaud-Trocmé et al., 2003). Further, synapses have a compressive nonlinearity for incoming spikes because of synaptic depression (Abbott et al., 1997). Here, we take $\log(x)$ to be the synaptic nonlinearity and $\exp(x)$ to be the spiking nonlinearity. For the aforementioned reasons, $u(t)$ can be interpreted quite literally as membrane potentials.

We define the model updates as $u_{t+1} = u_t + \delta u_t$, with $r_t = \exp(u_t)$ acting as the corresponding prior rates at time $t$, and $r_t \odot \delta r = \exp(u_{t+1})$, as the posterior rates at time $t$. When processing the next input in the sequence, we take the previous posterior and use it as our current prior. This works, because in the present paper, we restrict ourselves to stationary inputs comprised of the same image presented multiple times.

A natural choice for $\delta u$ is the gradient of the loss with respect to $u$, through the samples $z$. However, the KL term results in high order terms, which for this implementation we approximate as the following dynamics (See appendix D for a detailed derivation):

$$\delta u_t = J_\theta \cdot \Delta_t = \frac{\partial f_\theta(z)}{\partial z}\bigg|_{z=z_t} \cdot \big(x_t - f_\theta(z_t)\big),$$
(7)

where $J_\theta$ is the Jacobian of the decoder, $f_\theta$, which is a function of sampled spike counts $z$.

Importantly, this form aligns with real neuronal properties for several reasons. Since the comparison, $x_t - f_\theta(z_t)$, is based on spikes, each neuron's update does not directly depend on the internal states of other neurons, which matches how real neurons function (Kandel et al., 2000). Additionally, because the comparison happens on membrane potential (log rates), feedback will appear as a modulatory signal on rate, which is also consistent with neuroscience literature (Gilbert & Li, 2013). Finally, this update (eq. (7)) resembles a generalization of Rao & Ballard (1999) for nonlinear generative models and avoids hacky solutions to keep rates positive, after subtracting them.

It is straightforward to see how this is an SNN for linear decoder networks. If $f_\theta(z) = \Phi z$, then

$$\begin{aligned}
\delta u_t &= \Phi^T(x_t - \Phi z) \\
&= \Phi^T x - \Phi^T \Phi z \\
&= \Phi^T x - W z,
\end{aligned}$$
(8)

where the first term is the feedforward receptive fields (the input current) and the second term, $W$, are the recurrent weights between neurons, implementing lateral competition. Note that they only communicate with each other through spikes, $z$. Thus for linear generative models, i$\mathcal{P}$-VAE closely resembles the locally competitive algorithm for sparse coding (LCA; Rozell et al. (2008)), except that it is explicitly spiking and does not have a leak term (although this could included by replacing the diagonal of the recurrent term with a leak rather than having neurons operate on their own spikes).

In this section, we showed how following some fairly general assumptions for optimizing ELBO with Poisson distribution, led us to a spiking neural network that implements Bayesian posterior updates via predictive coding in the membrane potential dynamics. In the next section, we evaluate i$\mathcal{P}$-VAE and compare it to amortized $\mathcal{P}$-VAE, as well as iterative Gaussian VAEs.

## 4 EXPERIMENTS

We performed empirical analyses of i$\mathcal{P}$-VAE and alternative iterative VAE models. In section 4.1, we test the general performance and stability of inference dynamics, including generalization to longer sequence lengths. Section 4.2 shows i$\mathcal{P}$-VAE closes the gap with sparse coding. Section 4.3 demonstrates robustness to out-of-distribution (OOD) samples by evaluating models trained on MNIST (LeCun et al., 2010) with perturbed samples (e.g., rotated MNIST). We then evaluate OOD generalization from MNIST to other character-based datasets in section 4.3. Finally, in section 4.4, we visualize the learned weights of i$\mathcal{P}$-VAE, revealing their compositional nature, which is consistent with i$\mathcal{P}$-VAE's strong generalization capabilities. We push the limits of MNIST-trained models by testing their performance on natural images.

**Architecture notation.** We experimented with both convolutional and multi-layer perceptron (MLP) architectures. We highlight the encoder and decoder networks using red and blue, respectively. We use the ⟨enc|dec⟩ convention to clearly specify which type was used. For example ⟨mlp|mlp⟩ means both encoder and decoder networks were mlp. We use the notation ⟨jacob|mlp⟩ to denote our fully iterative (non-amortized) i$\mathcal{P}$-VAE. We chose symmetrical architectures, such that ⟨mlp|mlp⟩ has exactly twice as many parameters as ⟨jacob|mlp⟩.

**Datasets.** For the generalization results, we use MNIST, extended MNIST (EMNIST; Cohen et al. (2017)), Omniglot (Lake et al., 2015) and Imagenet32 (Chrabaszcz et al., 2017). We resize Omniglot and Imagenet32 to $28 \times 28$ for more straightforward comparisons. We also replicated the sparsity analysis in Fig. 3 of Vafaii et al. (2024) in our Table 1, using the van Hateren natural images dataset with whitened, contrast normalized $16 \times 16$ patches.

**Alternative models.** We compare our iterative $\mathcal{P}$-VAE (i$\mathcal{P}$-VAE) to $\mathcal{P}$-VAE. The main difference between their two architectures is that the latter independently parameterizes an encoder, whereas the former constructs its encoder adaptively by inverting the decoder. We also compare to state-of-the-art methods that combine iterative with amortized inference. These include iterative amortized VAE (ia-VAE; Marino et al. (2018)), and semi-amortized VAE (sa-VAE; Kim et al. (2018)). Since ia-VAE comes with both hierarchical (h) and single-level (s) variants, we compare to each of these.

**Number of iterations.** For i$\mathcal{P}$-VAE, we experimented with different numbers of training iterations, $T_{\text{train}}$. During training, we differentiate through the entire sequence of iterations, which can lead to qualitatively different dynamics. We report results for $T_{\text{train}} = 4, 16, 32, 64$. For generalization results, we use a model with $T_{\text{train}} = 64$. At test time, we report results using $T_{\text{test}} = 1,000$ iterations, unless stated otherwise. For semi-amortized models, we use their default number of train and test iterations found in their code, unless stated otherwise (sa-VAE: $T_{\text{train}} = T_{\text{test}} = 20$; ia-VAE: $T_{\text{train}} = T_{\text{test}} = 5$).

### 4.1 STABILITY BEYOND THE TRAINING REGIME AND CONVERGENCE.

An algorithm with strong generalization potential should learn how to perform inference that extends beyond the training regime. We evaluated this by training models on MNIST under different numbers of training iterations, $T_{\text{train}} = 4, 16, 32$, and $64$. We used both ⟨jacob|mlp⟩ and ⟨jacob|conv⟩ architectures and then tested each model on its ability to keep improving beyond the training number of iterations. In Fig. 1a, we show that i$\mathcal{P}$-VAE converges. Even with as few as 4 iterations, i$\mathcal{P}$-VAE learns to keep improving. We also observe that increasing the number of training iterations has an interesting effect: i$\mathcal{P}$-VAE trained with a larger number of iterations starts from worse performance, but converge to better solutions (Fig. 1a). This suggests i$\mathcal{P}$-VAE learns dynamics that depend on the training sequence length, but generalizes beyond the training set in all cases.

In contrast, the two hybrid models (sa-VAE and ia-VAE) start with strong amortized initial guesses, but plateau rapidly (Fig. 1a, right), and converge to a much higher MSE than i$\mathcal{P}$-VAE models, which

have a fraction of the parameters. The authors of sa-VAE were aware of issues regarding dominance of the iterative part of the algorithm for Omniglot, and reported using tricks like gradient clipping to mitigate it, which we suspect is the source for our observations on MNIST (see footnote 6 in Kim et al. (2018)). We also see that ia-VAE (single-level) starts to diverge outside its training regime. [1]

Overall, i$\mathcal{P}$-VAE achieves the best reconstruction performance and continues to improve outside the training regime, unlike other models. This shows the first sign of OOD generalization in i$\mathcal{P}$-VAE: temporal generalization. In later sections, we test whether i$\mathcal{P}$-VAE can generalize OOD in vision tasks, but first, we evaluate the performance and sparsity on natural images as in Vafaii et al. (2024).

### 4.2 IP-VAE CLOSES THE GAP WITH SPARSE CODING

One of the limitations of previous work with $\mathcal{P}$-VAE, was that the authors identified a large performance gap between $\mathcal{P}$-VAE and LCA sparse coding (Vafaii et al., 2024). Here, we evaluated i$\mathcal{P}$-VAE and compared models on their ability to reconstruct whitened natural image patches (table 1). Unlike $\mathcal{P}$-VAE, i$\mathcal{P}$-VAE performs as well as LCA with similar sparsity levels. $\mathcal{P}$-VAE, and the two hybrid approaches, have many more parameters and achieve much worse performance. [2]

---

[1]It's worth noting that in our hands, ia-VAE (s) often resulted in nans at test time upon going beyond $T_{\text{train}}$.

[2]The performance ia-VAE and sa-VAE might be modestly improved by tuning the tradeoff between reconstruction and the KL term.

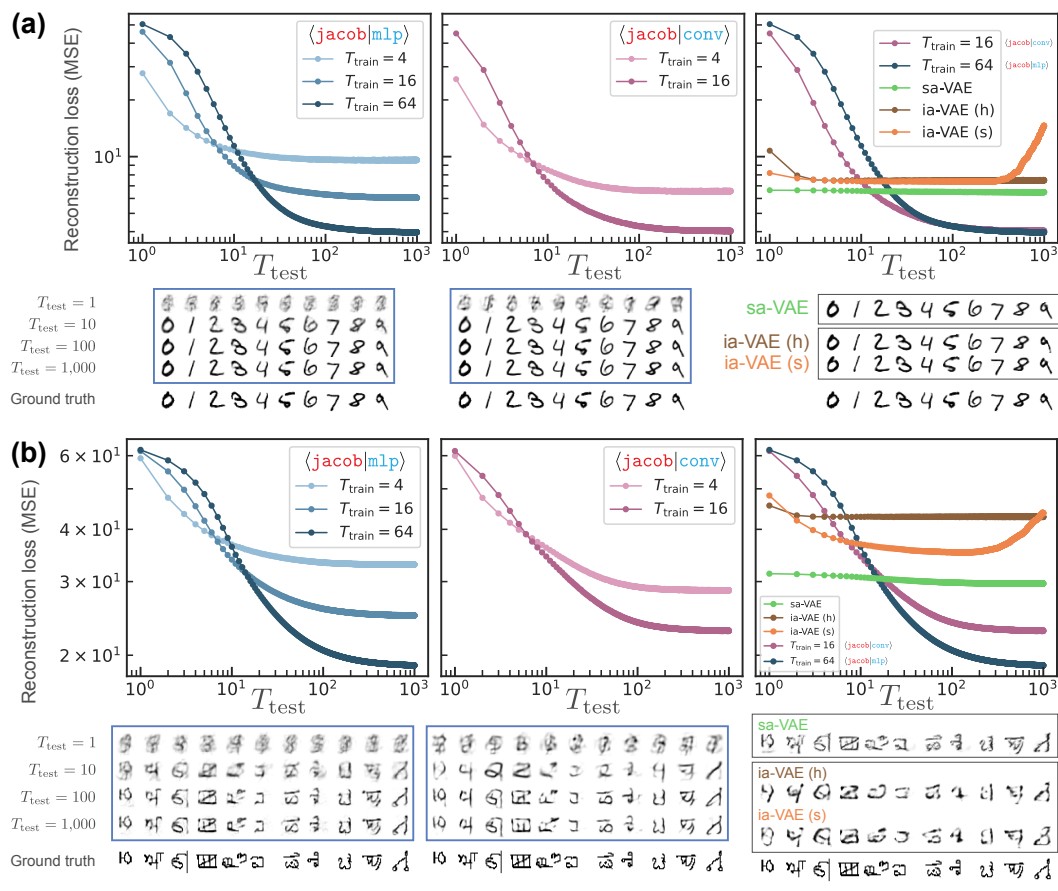

Figure 1: i$\mathcal{P}$-VAE learns to learn. **(a)** Training i$\mathcal{P}$-VAE on as few as $T_{\text{train}} = 4$ time steps allows it to generalize and keep improving its inference beyond the training domain. This holds true irrespective of the i$\mathcal{P}$-VAE architecture; left, $\langle\text{jacob}|\text{mlp}\rangle$; middle, $\langle\text{jacob}|\text{conv}\rangle$. In contrast, hybrid amortized/iterative models do not improve, and either remain flat or diverge (right). **(b)** i$\mathcal{P}$-VAE trained on MNIST generalizes to Omniglot at test time. All models in this figure were trained on MNIST, and tested either on MNIST (a), or Omniglot (b).

Table 1: Model performance and efficiency. We prefer lightweight models that achieve low reconstruction loss using sparse representations and fewer parameters. We reported results on natural image patches extracted from the van Hateren dataset (Van Hateren & van der Schaaf, 1998). All models have $K = 512$ dimensional latent space. For the i$\mathcal{P}$-VAE models, we scaled the $\beta$ parameter proportional to the number of training inference iterations. Specifically, we chose $\beta = 3/8 * T_{\text{train}}$. We found that i$\mathcal{P}$-VAE results were robust to variations in $\beta$. Entries formatted as mean±std.

| Model | $\beta$ | Architecture | # params ↓ | MSE ↓ | Sparsity ↑ | | # iters | |
| | | | | | lifetime | % | train | test |
|---|---|---|---|---|---|---|---|---|
| i$\mathcal{P}$-VAE | 24.00 | $\langle$jacob$\vert$lin$\rangle$ | 0.13 $M$ | **12.0**±2.6 | 0.79±.03 | 60.0 | 64 | $1K$ |
| i$\mathcal{P}$-VAE | 3.00 | $\langle$jacob$\vert$lin$\rangle$ | 0.13 $M$ | 27.5±7.1 | 0.85±.02 | 73.2 | 8 | $1K$ |
| i$\mathcal{P}$-VAE | 1.50 | $\langle$jacob$\vert$lin$\rangle$ | 0.13 $M$ | 50.4±15.5 | **0.90**±.03 | **83.3** | 4 | $1K$ |
| $\mathcal{P}$-VAE | 0.50 | $\langle$conv$\vert$lin$\rangle$ | 3.44 $M$ | 101.9±25.3 | 0.76±.16 | 65.9 | 1 | 1 |
| $\mathcal{P}$-VAE | 0.75 | $\langle$conv$\vert$lin$\rangle$ | 3.44 $M$ | 119.4±26.4 | 0.83±.09 | 77.7 | 1 | 1 |
| $\mathcal{P}$-VAE | 1.00 | $\langle$conv$\vert$lin$\rangle$ | 3.44 $M$ | 131.8±31.2 | **0.90**±.08 | **84.1** | 1 | 1 |
| LCA | 0.28 | - | 0.13 $M$ | **16.1**±8.1 | 0.79±.02 | 65.6 | $1K$ | $1K$ |
| LCA | 0.44 | - | 0.13 $M$ | 28.5±14.1 | 0.86±.02 | 73.9 | $1K$ | $1K$ |
| LCA | 0.70 | - | 0.13 $M$ | 50.1±25.2 | **0.92**±.01 | **83.4** | $1K$ | $1K$ |
| ia-VAE (s) | 1.00 | $\langle$mlp$\vert$mlp$\rangle$ | 39.55 $M$ | 80.08±21.06 | 0.36±.00 | ∼0.0 | 5 | 10 |
| sa-VAE | 1.00 | $\langle$conv$\vert$conv$\rangle$ | 1.67 $M$ | 97.74±38.97 | 0.36±.00 | ∼0.0 | 20 | 20 |

### 4.3 OUT-OF-DISTRIBUTION GENERALIZATION.

In this section, we evaluate whether MNIST-trained models generalize to OOD perturbations and dataset. First, we tested whether MNIST-trained models generalize to Omniglot (see Fig. 1b). We found that i$\mathcal{P}$-VAE improves over iterations and outperforms alternative models in terms of reconstruction quality. In this section, we evaluate two levels of generalization tasks: (1) within-dataset perturbations; and, (2) across similar datasets (i.e., digits to characters).

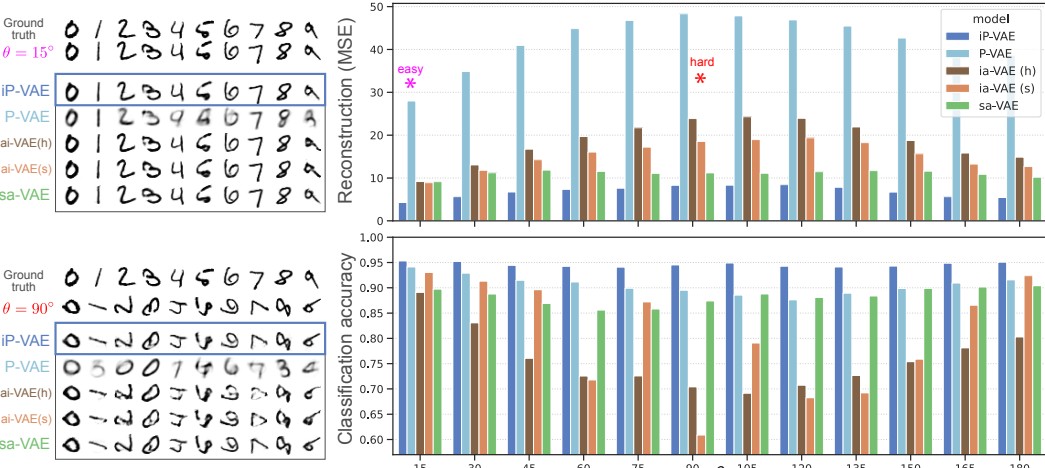

Figure 2: Robustness to training set perturbation. We rotated MNIST digits and evaluated model performance in both reconstruction of the perturbed inputs, and classification accuracy. On the left, we show reconstructed samples for easy ($\theta = 15°$) and hard ($\theta = 90°$) tasks across different models. On the right, we visualize the average reconstruction loss and classification accuracies over different rotations. Both visualy and quantitavely, i$\mathcal{P}$-VAE maintains a high performance regardless of the rotation, and outperforms alternative models.

**OOD generalization to within-dataset perturbation.** We tested whether models trained on standard MNIST generalized to rotated MNIST digits. We rotate MNIST between 0 and 180 degrees, with incremental steps of 15 degrees. We then test (a) whether models are capable of reconstructing the rotated digits, and (b) whether the representations of rotated digits can be used to classify them (Fig. 2). i$\mathcal{P}$-VAE and sa-VAE demonstrated consistent performance across angles, both in terms of reconstruction loss and classification accuracy. Amortized $\mathcal{P}$-VAE shows worse reconstruction performance than all iterative models, but its classification accuracy is remarkably consistent across angles, beating or matching all models except for i$\mathcal{P}$-VAE. ia-VAE variants were greatly affected by the rotation, with significant falloff in both their classification score and reconstruction. Overall, i$\mathcal{P}$-VAE maintains stable performance across rotations at levels above alternative models.

**OOD generalization across similar datasets.** If a model learns compositional features, and if it employs an effective inference algorithm that leverages those features, it should be able to represent datasets that are within the same distributional vicinity as the training set. To test this, we evaluated MNIST-trained models on EMNIST and Omniglot. We report both mean squared error (MSE) of reconstruction and classification accuracy [3].

Again, i$\mathcal{P}$-VAE exhibited superior reconstruction performance over other models, both visually and MSE (Fig. 3). It also had substantially higher classification accuracy, suggesting it learns a compositional code and has strong generalization potential.

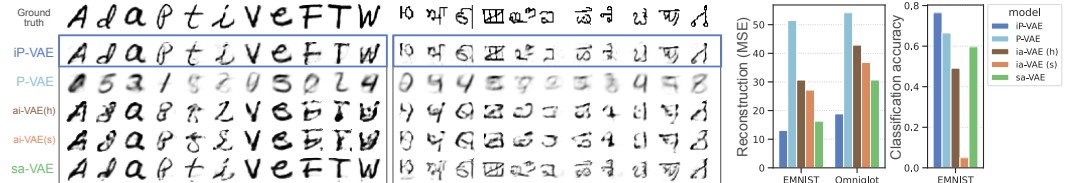

Figure 3: Evaluating generalization from models trained on MNIST digits to novel character datasets (EMNIST and Omniglot) at test time. The right panel shows the average classification performance on latent representations for EMNIST. The middle-right panel compares the reconstruction performance on EMNIST and Omniglot. The left two panels visualize the reconstructions on EMNIST and Omniglot, respectively. In both metrics, i$\mathcal{P}$-VAE maintains high performance compared to alternative models.

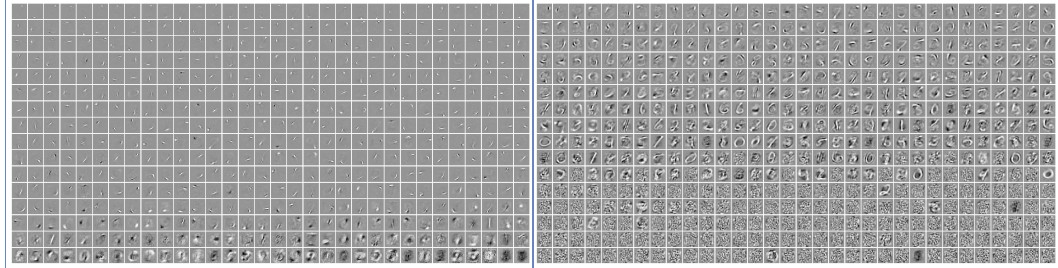

Figure 4: i$\mathcal{P}$-VAE learns a compositional set of features for the last layer's weights, enabling its generalization capacity. Left, i$\mathcal{P}$-VAE with a ⟨`jacob`|`mlp`⟩ architecture; right, $\mathcal{P}$-VAE with an ⟨`mlp`|`mlp`⟩ architecture. Both models were trained on MNIST, but only i$\mathcal{P}$-VAE develops Gabor-like features. In contrast, the non-iterative, amortized $\mathcal{P}$-VAE clearly overfits to MNIST. Features are ordered in ascending order of their weight distribution kurtosis to highlight the sparse nature of i$\mathcal{P}$-VAE feature space. Best viewed when zoomed in.

### 4.4 A COMPOSITIONAL CODE THAT GENERALIZES ACROSS DOMAINS .

Using the ⟨`jacob`|`mlp`⟩ variant of i$\mathcal{P}$-VAE, we visualized the 512 learned features of the last layer of the mlp decoder. In Fig. 4, we show the features learned by i$\mathcal{P}$-VAE trained on MNIST and contrast

---

[3]We omit classification accuracy for Omniglot due to its large number of classes (over 1,000)

them to features learned by $\mathcal{P}$-VAE, also trained on MNIST. We see a stark contrast. i$\mathcal{P}$-VAE features are Gabor-like, while $\mathcal{P}$-VAE features look like digits or strokes of the digits. While previous work highlighted strokes as the compositional subcomponents of digits (Lee et al., 2007), i$\mathcal{P}$-VAE learns an even more general code that generalized to cropped, grey scaled natural images (Fig. 5).

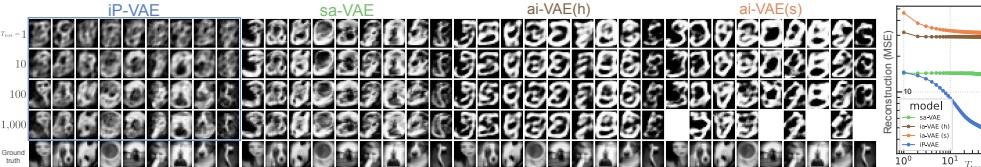

Figure 5: Evaluating generalization from models trained on MNIST digits to cropped, gray scaled natural images (ImageNet32) at test time. The right panel shows average reconstruction performance over inference iterations for the entire dataset. The left panels visualizes selected ground truth images compared with model reconstructions. The ai-VAE variants are unable to adapt to the new domain, whereas sa-VAE can capture more details. i$\mathcal{P}$-VAE outperforms the alternatives, and its reconstructions are shown to maintain the semantic information of ground truth images.

Since both i$\mathcal{P}$-VAE and $\mathcal{P}$-VAE are spiking models, this result suggests that the difference lies in the inference algorithm: i$\mathcal{P}$-VAE is iterative and adaptive; whereas, $\mathcal{P}$-VAE is one-shot amortized. Overall, our experiments provide strong evidence for the utility of iterative algorithms in practical settings.

## 5 DISCUSSION AND CONCLUSIONS

In this work, we introduce the i$\mathcal{P}$-VAE, which is a spiking neural network that maximizes ELBO, while performing Bayesian posterior updates through membrane potential dynamics. Empirically, i$\mathcal{P}$-VAE exhibits outstanding adaptability and robustness to OOD samples, while being able to dynamically trade off compute and performance. It outperforms amortized versions and recent iterative inference VAEs on every task we tested while using substantially fewer parameters.

i$\mathcal{P}$-VAE results directly from the choice of Poisson in the ELBO and it avoids many of the problems with predictive coding. First, there is no population-wide prediction signal, only a feedforward receptive field and recurrent terms. Second, neurons only communicate through spikes and all dynamics are private on the membrane potential. And finally, additive terms in the membrane potential appear as gains in the spike rate, which avoids negative rates, and is more consistent with real neurons Gilbert & Li (2013).

We believe i$\mathcal{P}$-VAE is well positioned for a neuromorphic implementation. The recent rise of neuromorphic hardware as an avenue for performance improvements requires new algorithms that can make use of its architecture (Schuman et al., 2022). We found that i$\mathcal{P}$-VAE with a linear decoder reduces to a spiking LCA, addressing the performance gap noted by Vafaii et al. (2024). Both algorithms share key features: sparsity, recurrence, and parameter efficiency. Since LCA has been implemented as an SNN (Zylberberg et al., 2011) and on neuromorphic hardware (Du et al., 2024), we expect the same for i$\mathcal{P}$-VAE.

In summary, the choice of Poisson in the ELBO results in a spiking neural network, i$\mathcal{P}$-VAE, that performs iterative Bayesian inference. This lays the groundwork for a prescriptive theoretical framework for building brain-like generative models that can leverage neuromorphic hardware.

**Limitations and future work.** In our experiments, we tested the simplest version of i$\mathcal{P}$-VAE, showing the practical benefits of the derived theory. There are a few avenues that we did not test, and we think are exciting for future work. The design of a hierarchical model is a natural extension for brain-like algorithm, especially given evidence that hierarchical VAE are more aligned to the brain (Vafaii et al., 2023). In addition, training and evaluating on nonstationary sequences like videos would be a straightforward extension, as we derived the theory with this in mind. When attempting to use such sequences, it may also be beneficial to explore more sophisticated forward-predictive models that "evolve" current posteriors to future priors.

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

## A EXPERIMENT DETAILS

In our comparisons to previous work, we utilized the code accompanied with sa-VAE (Kim et al. (2018)), ai-VAE (Marino et al. (2018)), and $\mathcal{P}$-VAE Vafaii et al. (2024). Across models where code was provided, we trained using the same train/validation split, and without changing the parameters in the code unless we specify otherwise. For the locally competitive algorithm (LCA) baseline, we used the library lca-pytorch (Teti, 2023) to replicate the analysis from Vafaii et al. (2024).

Since the code for sa-VAE was limited to a Bernoulli observation model, we adapted it for compatibility to Gaussian by removing the sigmoid in the decoder and replacing its reconstruction loss with MSE (for the van Hateren dataset). For sa-VAE, only Omniglot parameters were provided, with default batch size of 50, and default number of epochs of 100. We trained it on Omniglot with default parameters, on van Hateren for 100 epochs and batch size 200, on MNIST for 32 epochs and batch size 50, and EMNIST for 16 epochs and batch size 50, adjusting for the size and complexity of datasets.

The codebase for ai-VAE included parameters for both Bernoulli and Gaussian observation models, and we use them accordingly. We used their MNIST configuration for MNIST, EMNIST, and Omniglot. We used their CIFAR configuration for van Hateren, except for increasing batch size to 200 (van Hateren is much smaller spatially). For training the ai-VAE single-level model on van Hateren, we matched the latent dimension to all other van Hateren models (512 dims instead of 1024 from the CIFAR configuration). The number of epochs in the ai-VAE code base is hardcoded to 2000,

but we stopped the models between 780 and 2000 epochs when the loss converged. We found that the training code occasionally resulted in nans, requiring rerunning the training from the checkpoint. In one case, the hierarchical van Hateren model, the training was unable to proceed past 61 epochs without stopping due to nans.

We obtained the $\mathcal{P}$-VAE code upon request from the authors and used the default parameters as described in the appendix of Vafaii et al. (2024).

## B    ARE REAL NEURONS TRULY POISSON?

In this section, we discuss empirical and theoretical observations from neuroscience that support our Poisson assumption.

"Poisson-like" noise in neuroscience has a long history. It begins with observations that neurons do not fire the same sequence of spikes to repeated presentations of the same input and that the variance is proportional to the mean (Tolhurst et al., 1983; Dean, 1981) and was followed by the observation that for short counting windows, that proportion is 1 (Teich, 1989; Shadlen & Newsome, 1998; Averbeck et al., 2006; Rieke et al., 1999; Dayan & Abbott, 2005). Larger windows and higher visual areas are notably super-Poisson, but that can be attributed to a modulation of the rate of an inhomogeneous Poisson process (Goris et al., 2014). In other words, neurons are conditionally Poisson, not marginally Poisson (Truccolo et al., 2005).

Spike-generation, it is argued, is not noisy (Mainen & Sejnowski, 1995; Calvin & Stevens, 1968), but synaptic noise (Allen & Stevens, 1994) or noise on the membrane potential can create a Poisson-like distributions of spikes (Carandini, 2004). An important caveat is that the most famous examples of precision in spike generation, Mainen & Sejnowski (1995), is well captured well by a Poisson-process Generalized linear model (Weber & Pillow, 2017), although that precision depends on the Bernoulli approximation to a Poisson process in the limit where only 0 or 1 spikes are possible. There is a widely-held misconception that precise timing cannot be produced by spike-rate models, but inhomogeneous rate models can operate at high time resolution and produce precise spiking (Butts et al., 2016).

Importantly, to maximize the ELBO, one has to choose an approximate posterior and prior. Because spike counts are integer and cannot be negative, Poisson is a more natural choice than Gaussian without knowing anything about neural firing statistics. Here, we found that Poisson assumption produced a prescriptive theory for neural coding. Future work might interpret this assumption at higher time resolution using inhomeneous Poisson processes in the limit of binary spiking.

## C    EXTENDED RELATED WORKS

### C.1    DIFFUSION MODELS

Diffusion models have recently gained significant traction in various generative tasks, demonstrating impressive performance across applications (Yang et al., 2024; Chan, 2024). Originally introduced by Sohl-Dickstein et al. (2015), these models iteratively restore data structure by learning a reverse diffusion process. Despite the dominance of one-shot feedforward methods, the success of diffusion models highlights the ongoing relevance of iterative approaches. Several studies have sought to explain why these models perform so well in tasks like image generation. In this section, we highlight three key findings.

First, Delbracio & Milanfar (2024) and Bansal et al. (2022) showed that fully deterministic iterative restoration methods, without diffusion theory, can match the performance of conditional diffusion models. This suggests that the strength of diffusion models lies, at least partially, in their iterative nature.

Second, Kingma & Gao (2023) revealed that despite their distinct loss functions, diffusion models essentially optimized the ELBO objective (identical under certain conditions), particularly in noise-perturbed data settings. This adds further support to the idea that diffusion models succeed not because of their diffusion-specific properties, but because they are iterative, aligning them closely with i$\mathcal{P}$-VAE, which also optimizes an ELBO-like objective through iterative processes.

Finally, Kadkhodaie et al. (2024) found that diffusion models operate by applying a shrinkage operation on an adaptive basis, a fundamental concept in signal processing. In methods like sparse coding, this is represented by an $L1$ regularization term. Similarly, an $L1$-like term appears in i$\mathcal{P}$-VAE, which also uses integer representations to zero out small values. These similarities suggest a strong connection between i$\mathcal{P}$-VAE and diffusion models, presenting an exciting direction for future research.

## C.2 ADAPTIVE FILTERS

Adaptive filters are a widely used class of algorithms capable of modeling signals with varying statistics (Widrow & Stearns (1985)). Their applications are highly diverse, including communications, control and robotics, weather prediction, and inverse problems such as denoising. Two of the most popular adaptive filter classes, the Kalman filter (Kalman (1960)) and the Least mean squares (LMS) filter (Widrow & Stearns (1985)), have close connections to machine learning. The LMS filter was originally based on research aiming to train neural networks (Widrow (1960)). Backpropagation can be understood as a generalization of the LMS filter when applied to multi-layer networks. Although the Kalman filter has not had much use as a learning algorithm, a recent line of work shows that there is a lot of potential benefits in doing so (Trautner et al. (2020); Luttmann & Mercorelli (2021)). Both algorithms, when used in dynamic settings, encode the prediction residual (like i$\mathcal{P}$-VAE), and can be interpreted from the framework of predictive coding. More concretely, Millidge et al. (2021a) showed predictive coding in the linear case corresponds to Kalman filtering, and also showed the relationship between backpropagation (extension of LMS) and predictive coding. Later, Millidge et al. (2021b) showed that predictive coding and Kalman filtering, although not identical in general, optimize the same objective. In addition, they show a neurally plausible implementation of the Kalman filter (see Wilson & Finkel (2009) for an earlier paper in this line of work).

In future work, it would be interesting to incorporate additional ideas from the rich literature of Kalman filters. Particularly, extensions of Kalman filtering, such as the ensemble Kalman filtering, tend to be better suited for nonlinear and nongaussian applications (albeit with the loss of guarantees).

## C.3 TEST-TIME OPTIMIZATION

There has been a recent surge of work showing that incorporating test-time optimization leads to improved performance. One notable line of work is known as Test-Time-Training (TTT), introduced by Sun et al. (2020). TTT is a general approach for updating model parameters in test time using self-supervised learning, demonstrating increased performance and robustness. Around the same time Quan et al. (2020) introduced Self2Self, a denoising method that is only trained during test time. A follow-up to Self2Self instead optimized a per-layer gain value of a trained model Mohan et al. (2021). In a recent paper, Sun et al. (2024) extended the TTT framework to language modeling, introducing an architecture that outperforms transformers (Vaswani et al., 2017) and Mamba (Gu & Dao, 2023). The authors also showed that theoretically, transformers can be understood as a special case of their TTT algorithm. In this work, we found that i$\mathcal{P}$-VAE can also be understood within the TTT framework. Overall, our results reveal a novel grounding of TTT within well-established theoretical concepts in neuroscience.

## C.4 FEEDFORWARD VERSUS ITERATIVE COMPUTATION

Deep learning is currently the dominant paradigm in artificial intelligence (AI) research, driven largely by the success of feedforward neural networks (LeCun et al., 2015; Sejnowski, 2020). The deep learning era invoked the universal approximation theorem (Hornik et al., 1989) and emphasized parallelization of training (Krizhevsky et al., 2012; Vaswani et al., 2017) leading to an over-reliance on models that perform one-shot inference. This "unrolling" of inference diverged from the classic AI literature, which recognized the importance of iterative algorithms (Russell & Norvig, 2016). Although feedforward models initially achieved remarkable results, their limitations became increasingly apparent as they struggled to generalize beyond their training distributions (Zhou et al., 2022; Yu et al., 2024). To counter this limitation, iterative computation at test time has recently resurfaced as a promising direction (Sun et al., 2020; 2024).

Unlike feedforward models, iterative algorithms refine their predictions over multiple steps, allowing them to adapt dynamically to new inputs. Examples include iterative amortized inference techniques Marino et al. (2018); Kim et al. (2018), diffusion models Sohl-Dickstein et al. (2015); Ho et al. (2020); Song & Ermon (2019), energy based models (Du & Mordatch, 2019; LeCun et al., 2006), test-time training Sun et al. (2020; 2024), meta-learning algorithms (Andrychowicz et al., 2016; Finn et al., 2017; Hospedales et al., 2021), neural ordinary differential equations (Chen et al., 2018), deep equilibrium models (Bai et al., 2019; 2020), object-centric models (Locatello et al., 2020; Chang et al., 2022), and many more. These methods have demonstrated that a dynamic, multi-step inference process can help overcome many of the challenges faced by static models.

### C.5 FAST WEIGHTS

In the late 1980s and early 1990s, Hinton & Plaut (1987) and Schmidhuber (1992) introduced the concept of "fast weights" as a way to enhance the adaptability of neural networks through dynamic memory. These innovations laid the foundation for modern models like transformers and recurrent neural networks, significantly influencing memory-augmented architectures and iterative inference methods. Fast weights are particularly relevant in iterative inference, where dynamic updates align with the goal of flexible, adaptive neural computation (Ba et al., 2016; Irie et al., 2021). In our work, the adaptive Bayesian posterior updates in $\boldsymbol{u}(t)$—the membrane potential state of i$\mathcal{P}$-VAE—closely parallel the concept of fast weights.

## D DYNAMICS

In this section, we will go through the derivation of the dynamics of i$\mathcal{P}$-VAE ( eq. (7) in the main paper). Our goal is to define membrane potential updates in a way that the resulting dynamics will minimize the ELBO loss.

We begin with the general definition of the ELBO, $\mathbb{E}_{q_\phi(\boldsymbol{z}|\boldsymbol{x})}\left[\log \frac{p(\boldsymbol{x},\boldsymbol{z})}{q_\phi(\boldsymbol{z}|\boldsymbol{x})}\right]$, and consider its Monte Carlo estimate using a single sample, $\boldsymbol{z}$, drawn from the approximate posterior $q_\phi(\boldsymbol{z}|\boldsymbol{x})$:

$$
\begin{aligned}
\ell(\boldsymbol{x}, \boldsymbol{z}) &:= \log \frac{p(\boldsymbol{x}, \boldsymbol{z})}{q_\phi(\boldsymbol{z}|\boldsymbol{x})} \\
&= \log \frac{p(\boldsymbol{x}|\boldsymbol{z})p(\boldsymbol{z})}{q_\phi(\boldsymbol{z}|\boldsymbol{x})} \\
&= \log p(\boldsymbol{x}|\boldsymbol{z}) + \log \frac{p(\boldsymbol{z})}{q_\phi(\boldsymbol{z}|\boldsymbol{x})} \\
&= -\mathrm{MSE}(\boldsymbol{x}, \boldsymbol{z}) + \boldsymbol{r} \odot (\exp(\boldsymbol{\delta u}) - 1) - \boldsymbol{z} \odot \boldsymbol{\delta u}.
\end{aligned}
\tag{9}
$$

In the last line of eq. (9), we inserted our specific choice of Gaussian conditional density, resulting in $\log p(\boldsymbol{x}|\boldsymbol{z}) = -\mathrm{MSE}(\boldsymbol{x}, \boldsymbol{z}) = -\|\boldsymbol{x} - f_\theta(\boldsymbol{z})\|^2$. We also expressed the log ratio between the prior and approximate posterior distributions, both modeled as Poisson, as in the case in i$\mathcal{P}$-VAE.

Next, we take the partial derivative of $\ell(\boldsymbol{x}, \boldsymbol{z})$ w.r.t the samples $\boldsymbol{z}$ and keep only the first order terms. This results in:

$$
\frac{\partial}{\partial \boldsymbol{z}}\ell(\boldsymbol{x}, \boldsymbol{z}) \approx -\frac{\partial}{\partial \boldsymbol{z}}\mathrm{MSE}(\boldsymbol{x}, \boldsymbol{z}) - \boldsymbol{\delta u}.
\tag{10}
$$

If we define our posterior updates, $\boldsymbol{\delta u}$, to be proportional to the gradient of $\ell(\boldsymbol{x}, \boldsymbol{z})$ w.r.t the state variable, $\boldsymbol{u}$, we get:

$$\boldsymbol{\delta u} := \alpha \nabla_{\boldsymbol{u}} \ell(\boldsymbol{x}, \boldsymbol{z})$$

$$= \alpha \frac{\partial \boldsymbol{z}}{\partial \boldsymbol{u}} \frac{\partial}{\partial \boldsymbol{z}} \ell(\boldsymbol{x}, \boldsymbol{z}) \tag{11}$$

$$\approx -\alpha \frac{\partial \boldsymbol{z}}{\partial \boldsymbol{u}} \left[ \frac{\partial}{\partial \boldsymbol{z}} \mathrm{MSE}(\boldsymbol{x}, \boldsymbol{z}) + \boldsymbol{\delta u} \right],$$

where $\alpha$ is a proportionality constant. We rearrange some terms to get the following update rule:

$$\boldsymbol{\delta u} = -\left( \frac{\alpha \partial \boldsymbol{z}/\partial \boldsymbol{u}}{1 + \alpha \partial \boldsymbol{z}/\partial \boldsymbol{u}} \right) \frac{\partial}{\partial \boldsymbol{z}} \mathrm{MSE}(\boldsymbol{x}, \boldsymbol{z}). \tag{12}$$

The stochastic samples, $\boldsymbol{z}$, depend to the state variable, $\boldsymbol{u}$, through firing rates, $\boldsymbol{r} = \exp(\boldsymbol{u})$. Therefore, we have $\partial \boldsymbol{z}/\partial \boldsymbol{u} = (\partial \boldsymbol{z}/\partial \boldsymbol{r})(\partial \boldsymbol{r}/\partial \boldsymbol{u})$. But $\partial \boldsymbol{r}/\partial \boldsymbol{u}$ is just $\boldsymbol{r}$, and if we approximate $\partial \boldsymbol{z}/\partial \boldsymbol{r}$ using the straight-through estimator, we will have $\partial \boldsymbol{z}/\partial \boldsymbol{u} \approx \boldsymbol{r}$. Plug this back into eq. (12) to get:

$$\boldsymbol{\delta u} \approx -\left( \frac{\alpha \boldsymbol{r}}{1 + \alpha \boldsymbol{r}} \right) \frac{\partial}{\partial \boldsymbol{z}} \mathrm{MSE}(\boldsymbol{x}, \boldsymbol{z}). \tag{13}$$

The proportionality coefficient, $\alpha \boldsymbol{r}/(1 + \alpha \boldsymbol{r})$, can be interpreted as an adaptive learning rate that depends on the instantaneous firing rate of neurons. While this result is intriguing, in the present work we simplified our update rule by removing the proportionality coefficient. Instead, we simply used the gradient of the MSE to compute $\boldsymbol{\delta u}$:

$$\boldsymbol{\delta u} \propto -\frac{\partial}{\partial \boldsymbol{z}} \mathrm{MSE}(\boldsymbol{x}, \boldsymbol{z})$$

$$= -\frac{\partial}{\partial \boldsymbol{z}} \|\boldsymbol{x} - f_\theta(\boldsymbol{x})\|^2 \tag{14}$$

$$\propto \frac{\partial f_\theta(\boldsymbol{z})}{\partial \boldsymbol{z}} \cdot \left( \boldsymbol{x}_t - f_\theta(\boldsymbol{z}_t) \right)$$

$$= \boldsymbol{J}_\theta \cdot \Delta_t.$$

This concludes our derivation of eq. (7).

