# OpenReview forum: "A prescriptive theory for brain-like inference"
_ICLR.cc/2025/Conference — Submitted to ICLR 2025_

### Official Review · Reviewer_MP3g · 2024-10-30

**Soundness:** 2
**Presentation:** 2
**Contribution:** 2
**Rating:** 3
**Confidence:** 3

**Summary:**

The authors introduce a new Variational Auto Encoder Architecture based on Poisson distribution and iterative inference (iP-VAE) and connect their new algorithm to sparse coding. They derive an Evidence lower bound for a sequence of poisson distributed latent encodings.
They evaluate the utility of their model in terms of convergence, efficiency, and out of distribution generalisation.

**Strengths:**

The topic is of interest. The goal of matching Machine learning concepts to the neuroscience literature is important.
The related work (section 2) is, in general, well written and structured.

**Weaknesses:**

I have, unfortunately, a hard time understanding most parts of the experimental section and what the authors aim to convey.
Variational Auto Encoders (VAEs) are generative models. The loss they are (mostly) trained with (which derives from the ELBO) has two components, a reconstruction-term and a regularisation-term. The reconstruction-term typically refers to the term in the ELBO encapsulating the expected conditional log likelihood of the observation given the latent variable. The regularisation-term typically measures the KL-divergence between the posterior and the prior distribution.

In their experimental section, the authors chose to merely measure the reconstruction performance, i.e. only part of the loss function.
Generally, by putting small weights on the regularisation term (KL-Divergence) a VAE becomes better in reconstruction but loses its generative capabilities. The works the authors compare with, semi amortised VAE [1] and amortised VAE [2] generally report either the actual ELBO loss function or conditional loglilekelihood (reconstruction) together with KL-Divergence (regularisation). They are also both six years old but have reported on datasets of higher complexity than the authors of the paper at hand.

[1] Kim, Yoon, et al. "Semi-amortized variational autoencoders." International Conference on Machine Learning. PMLR, 2018.
[2] Marino, Joe, Yisong Yue, and Stephan Mandt. "Iterative amortized inference." International Conference on Machine Learning. PMLR, 2018.

**Questions:**

Please refer to the Weaknesses section for questions and concerns.
I suggest the authors try to compare to more recent (and peer-reviewed) papers. I also suggest that the authors either illustrate why only comparing reconstruction is reasonable, or align their work more closely with the existing literature by a more comprehensive comparison between models (reflecting both regularisation and reconstruction)

---

> ### Author Response · Authors · 2024-11-26
> **Clarifying iP-VAE's priorities: Inference over generative performance**
>
> Thank you for your thoughtful comments. Below, we address the key points you raised and provide clarifications that we believe will improve the presentation of our work.
>
> > Variational Auto Encoders (VAEs) are generative models… Generally, by putting small weights on the regularisation term (KL-Divergence) a VAE becomes better in reconstruction but loses its generative capabilities.
>
> This comment, and another comment from Rev. JGdX, made us realize that we did not clearly communicate our intentions in the introduction. With iP-VAE, our primary goal was to introduce a powerful and adaptive inference algorithm, and generative modeling performance was not our top priority. That said, we performed some preliminary experiments and found that, at least on MNIST, iP-VAE generates higher-quality images compared to ia-VAE and sa-VAE.
>
> > In their experimental section, the authors chose to merely measure the reconstruction performance, i.e. only part of the loss function.
>
> This is a valid point. We motivated our model for general sequences, but we only evaluated on a single image repeated many times to compare to existing iterative VAE models. Due to this conceptual difference, the comparison across models for the KL term becomes complicated, which is why we only focused on the reconstruction term.
>
> In the future version of the paper, we plan to report the full ELBO or importance-weighted log-likelihood measures and include comparisons with more recent sequence latent variable models.

---

### Official Review · Reviewer_JGdX · 2024-11-03

**Soundness:** 2
**Presentation:** 3
**Contribution:** 2
**Rating:** 5
**Confidence:** 3

**Summary:**

The paper shows that iterative Poisson VAE can be a reasonable prescriptive model for biological neural networks. Compared to Poisson VAE, iterative nature of the IP-VAE reduces the amortization gap, leading to superior reconstruction performance, sparse features, and OOD generalization.

**Strengths:**

The paper is well-motivated, and the experiments are extensive.

Interesting results on the sparse features learned by iP-VAE is shown.

**Weaknesses:**

My biggest concern for this paper is that the authors did not show the generative performance of IP-VAE, which is important for any Bayesian model of the brain. Also, if I understand the paper correctly (line 242-245), at each time point the IP-VAE receives the ground truth image, making the superior reconstruction performance, generalization and parameter efficiency less impressive. For generative models, the info about the data distribution is usually stored in the parameters, it is natural that iP-VAE does not need many parameters if the ground truth samples are provided.

**Questions:**

1. In Figure 4 towards the bottom row of features learned by iP-VAE, it seems some features also overfit to digit-like shape. I wonder if iP-VAE can achieve better generalization performance without these features (by deleting those overfitted features and freezing the rest of the features)

2. What is $\beta$ in Table 1?

---

> ### Author Response · Authors · 2024-11-26
> **Inference is our priority, generative capacity comes secondary**
>
> Thank you for your thoughtful comments. We’ve responded below to clarify our intentions and address your concerns, which we believe will strengthen future iterations of our work.
>
> > My biggest concern for this paper is that the authors did not show the generative performance of IP-VAE…
>
> This comment, and another comment from Rev. MP3g, made us realize that we did not clearly communicate our intentions in the introduction. With iP-VAE, our primary goal was to introduce a powerful and adaptive inference algorithm, and generative modeling performance was not our top priority. That said, we performed some preliminary experiments and found that, at least on MNIST, iP-VAE generates higher-quality images compared to ia-VAE and sa-VAE.
>
> > …which is important for any Bayesian model of the brain
>
> Here we disagree. It is true that in our work, we start with the assumption that the brain contains a generative model of the world. This is a useful assumption for model building, but we do not think the brain’s generative model should be capable of generating high-quality images at pixel level by only drawing from the prior, without any prompting, as is the standard approach in VAEs. In our view, adaptive inference is more fundamental, which in our case, utilizes some generative capacity. Technically, our algorithm performs *analysis-by-synthesis* ([Yuille & Kersten, 2006](https://www.sciencedirect.com/science/article/abs/pii/S1364661306001264)), where the *synthesis* part is powered by the generative model. But the ultimate goal is *analysis* or inference.
>
> This requires a much deeper discussion, but we plan to include something like the paragraph above in the future version of our paper for clarity.
>
> > …at each time point the IP-VAE receives the ground truth image, making the superior reconstruction performance, generalization and parameter efficiency less impressive.
>
> This is also true for amortized algorithms, as they receive the ground truth image to be reconstructed. The difference is that in amortized settings, inference is typically done in a one-shot manner, while our algorithm performs inference iteratively.
>
> > In Figure 4 towards the bottom row of features learned by iP-VAE, it seems some features also overfit to digit-like shape. I wonder if iP-VAE can achieve better generalization performance without these features (by deleting those overfitted features and freezing the rest of the features)
>
> This is an interesting point. We plan to explore this in the future version of the paper.
>
> > What is $\beta$ in Table 1?
>
> It is a parameter that determines the trade-off between reconstruction and the KL term (like beta-VAE). For LCA, it simply sets the strength of the L1 regularization.
>
> We greatly value your feedback and would appreciate any further insights you might share in response to our comments. Thank you again for your time and thoughtful review.

---

> ### Comment · Reviewer_JGdX · 2024-12-03
>
> I am confused. In the first paragraph the authors claim that iP-VAE generates high quality images, but then in the second paragraph they claim that "the brain's generative model should [not] be capable of generating high-quality images by only drawing from the prior". How does the author generate the images then?
>
> The brain also does not necessarily generate the image directly, it could simply generate high-quality latent representation that is decoded by FFNs into "high-quality images at pixel level", very much like what latent diffusion (Rombach et al., 2021) does. Speaking of which, recent work has already begun to look at how neural circuits can sample like a diffusion model [2,3], therefore I don't think the authors' argument that "brain cannot generate high quality image" holds. Finally, there is this neurological phenomenon called hyperphantasia that directly disprove the author's argument.
>
> Nevertheless, I thank the authors for bringing up this topic, and I enjoy the discussion.
>
> [1] Rombach, Robin, et al. "High-resolution image synthesis with latent diffusion models." Proceedings of the IEEE/CVF conference on computer vision and pattern recognition. 2022.
>
> [2] Lyo, Benjamin SH, and Cristina Savin. "Complex priors and flexible inference in recurrent circuits with dendritic nonlinearities." bioRxiv (2023): 2023-10.
>
> [3] Chen, Shirui, et al. "Expressive probabilistic sampling in recurrent neural networks." Advances in Neural Information Processing Systems 36 (2024).

---

### Official Review · Reviewer_u291 · 2024-11-04

**Soundness:** 3
**Presentation:** 3
**Contribution:** 2
**Rating:** 6
**Confidence:** 3

**Summary:**

This paper proposes iterative Poisson VAE (iP-VAE) which maximizes ELBO for sequence Poisson observations. The authors demonstrate the superior generalization to out of distribution samples of the proposed method by evaluating on a few real datasets.

**Strengths:**

- The paper is clearly written.
- The idea of drawing connection to membrane potential dynamics is interesting.
- The authors have done extensive experiments on a few real datasets, and the results seem good.

**Weaknesses:**

- I listed some clarification questions in the section below.

**Questions:**

- Intuitively, why modeling the latent variable z_t as Poisson distribution is better than choosing it as Gaussian as in the vanilla VAE?
- In the experiment section, T_test for iP-VAE is much higher than ia-VAE and sa-VAE. What's the running time comparison for training and inference on test samples?

---

> ### Author Response · Authors · 2024-11-26
> **Addressing reviewer's questions on Poisson latents and run-time metrics**
>
> Thank you for your feedback and the questions you raised. Below, we provide brief responses to your points.
>
> > Intuitively, why modeling the latent variable z_t as Poisson distribution is better than choosing it as Gaussian as in the vanilla VAE?
>
> Our primary motivation is that Poisson makes representations more brain-like, as the brain uses discrete spike counts to represent and communicate information. An ANN model using similar representations would enable in-silico experimentation, offering insights into neural computation principles that might transfer from the ANN to the brain. Without a similar representational form, this would be much more challenging.
>
> There are other benefits, too. Apart from the out-of-distribution generalization results reported in our submission, it was shown that, for example, Poisson latents largely alleviate the prevalent posterior collapse issue in VAEs, leading to much fewer dead neurons (See Figures 4 and Table 2 in [Vafaii et al., 2024](https://openreview.net/forum?id=ektPEcqGLb)).
>
> > In the experiment section, T_test for iP-VAE is much higher than ia-VAE and sa-VAE. What's the running time comparison for training and inference on test samples?
>
> While we did not quantify this, we agree that it is an important metric to report. In the future, we aim to reduce T_test by including momentum terms, and will rigorously quantify and report run-time metrics.

---

### Official Review · Reviewer_hNnm · 2024-11-04

**Soundness:** 2
**Presentation:** 2
**Contribution:** 2
**Rating:** 3
**Confidence:** 3

**Summary:**

Variational autoencoders (VAE) are useful types of generative models, but they are not generally brain-like. In particular, they do not typically operate with spikes, and they are not typically 'dynamic' (beyond the single encoding-decoding pass). The authors introduce a novel type of VAE that involves Poisson (and hence spike-like) latent variables and iterative updating. They claim that this VAE, the iP-VAE, exhibits a number of nice properties, including sparse representations and a "compositional code" that supports performant generalization. They also relate various aspects of the iP-VAE to well-known frameworks from neuroscience, including sparse coding, predictive coding, and the Free Energy Principle (FEP), and claim that it may provide a "solid foundation" for further work in neuroAI.

**Strengths:**

The core idea of the work, which is to modify VAEs to have a latent space that is dynamic and more neurally plausible, is interesting and seems to lead to nontrivial performance gains, at least on the data sets the authors tested.

The authors do a nice job reviewing relevant literature and relating their proposal to existing frameworks and models from neuroscience.

**Weaknesses:**

My major concerns are related to (1) experimental support for claims about the iP-VAE, (2) technical details related to the iP-VAE, and (3) the abundance of wildly overstated claims, particularly related to how the authors' contribution is framed. As a minor concern, there are some citation typos (e.g., line 521, no parentheses).

**1. Experimental support.** As far as I can tell, the authors only trained iP-VAEs on MNIST, and checked its out-of-distribution generalization performance on a few other data sets (including extended MNIST and ImageNet-32). Given the grandiosity of some of the claims (see below), it seems reasonable to expect results for more and/or more complex data sets. In order to contextualize the performance of iP-VAEs, the authors only compare it to very similar models, including the recently introduced P-VAE; it would be extremely helpful to also compare its performance to the vanilla VAE and other types of generative models, like diffusion models (which the authors mention). If the iP-VAE does not perform as well, and its benefits are better thought of as related to its brain-like properties, the authors should state this explicitly.

In Sec. 4.4, the authors claim that the iP-VAE learns a "compositional code that generalizes across domains". This is a really strong claim, and is very interesting if true, but only some sample reconstructions and reconstruction loss performance (Fig. 5) are shown to back it up. At least to me, such a claim requires some quantitative analysis of the learned representations. If the authors cannot do such an analysis or consider it out of scope, they should modify the claim to something like "suggests the existence of a compositional code", and note that verifying this would require additional work analyzing representations.

**2. Technical details regarding iP-VAE.** First, there is the high-level concern that the iP-VAE is pretty similar (in terms of details and motivation) to the recently introduced P-VAE, so there is a novelty issue. Second, I am slightly confused by the way certain details of the construction are motivated and presented in Sec. 3.

The authors claim (line 176-177) assuming a Markovian relationship between consecutive data points is a reasonable starting point for modeling the world, but this seems wrong. What people usually do is assume that *latent variables* are Markovian, and that observations are generated from those latent variables. It is obvious that observations should not be treated as Markovian: how the current scene in front of me changes strongly depends on what is happening in the world (most of which I can't see at that moment), and is not usually even approximately Markovian.

Related to the previous point, I would have expected instead an assumption that the $z_t$ variables are Markovian, and that the $x_t$ variables are viewed as 'noisy' observations of them. In general, I find the structure of the generative model kind of confusing (a schematic figure to illuminate this would help). Also, even though the 'iterative' aspect of the iP-VAE is presented as the main novel contribution (at least to my knowledge), on all inputs the authors consider there is no dynamics; instead, the same input is presented to the model many times.

The authors claim that the iP-VAE performs Bayesian posterior updates using membrane potential dynamics, but this seems like an overstatement. Usually membrane potential dynamics involve some kind of spiking threshold rule (e.g., consider leaky integrate and fire neurons), and there is no such rule here. It is more accurate to say that updates occur in log-rate space, and that this is 'membrane-potential-like' somehow. Even though the authors mention the possibility of including a leak term to make their model more neurally plausible, they do not include one in any experiments.

**3. Grand claims.** Perhaps the biggest issue with this paper, beyond the need for more experimental support, regards the grandiose claims throughout. The title of the paper is "A prescriptive theory for brain-like inference", but the authors' theory is neither prescriptive nor particularly brain-like, or at least it is not especially *more* brain-like than other models that have been proposed.

The word "prescriptive" suggests that the authors demonstrate how one *ought* to do brain-like inference. But this is not what they showed. They showed that a specific modification of a VAE is slightly brain-like and performs decently well when trained on MNIST. There are many other ways of doing brain-like inference: for example, a NeurIPS paper I am familiar with (Masset and Zavatone-Veth et al. 2022, "Natural gradient enables fast sampling in spiking neural networks") discusses ways based on Langevin and MCMC sampling. Is the proposed approach substantially better (along some axis of 'brain-like-ness') than other approaches? This is not clear from the paper, since existing approaches also involve membrane-potential-like dynamics and sampling from some kind of latent space. Noting difficulties with other approaches, as in done in a comparison to predictive coding on line 63, would be helpful. Incidentally, saying predictive coding doesn't involve spikes is a little unfair, considering forms of it exist (to my knowledge) that address this issue.

The term "brain-like" here may be a bit overstated. What the authors propose is more brain-like in certain ways (e.g., Poisson distributions are used in the latent space, and representations are relatively sparse), but the extent of brain-like-ness in other ways is unclear (e.g., Is the proposed update rule, described by Eq. 7 and derived in Appendix D, neurally plausible? This depends somewhat on how the Jacobian is computed or approximated.). As a related point, real neurons are only approximately Poisson, and only sometimes. It is probably *more* brain-like to consider them than something like a Gaussian, but it's still far from an actual brain. Also (line 236), an exponential is not the only (or obviously best) way to model the spike threshold of real neurons, at least to my knowledge.

The authors claim that their work provides a "solid foundation for developing prescriptive theories in NeuroAI", but this seems overly strong. If I want to solve problem X subject to a set of constraints Y, how does the iP-VAE work help me? Only a narrow problem setting is considered here compared to the galaxy of problems and constraints relevant for neuroAI.

The authors claim that the iP-VAE is particularly well-suited to neuromorphic implementations (paragraph starting on line 522), but this seems premature given the content of the paper.

Overall, many big claims regarding how the iP-VAE solves fundamental problems, or is prescriptive, or is particularly well-suited to some future thing, should probably be tamped down, and more time should be spent on supporting the claim that it works well and is especially (along some axes) brain-like.

**Questions:**

1. Can the authors clarify the structure of their proposed generative model? Why are observations assumed to be Markov, and what real-world assumption does this relate to?
2. Does iP-VAE perform well compared to vanilla VAEs and other types of generative models, like diffusion models?
3. Can the iP-VAE be trained on a more complex data set than MNIST? If there are technical difficulties, what are they?
4. In what sense does the iP-VAE learn a "compositional code"? Can the authors show representation analysis results to support this?
5. How would a leak term affect the iP-VAE?
6. How is the iP-VAE more brain-like than other proposed brain-like inference approaches? See the earlier Masset and Zavatone-Veth et al. citation, although there are many more such examples (e.g., recent modifications of predictive coding), and the specific paper isn't that important. More comparisons here would be really helpful.
7. How is the authors' proposal "prescriptive"?

---

> ### Author Response · Authors · 2024-11-26
> **Clarifications and future directions in response to reviewer's feedback (1/2)**
>
> Thank you for your detailed and thoughtful comments. They have helped us identify important areas for improvement. Below, we address some of the key points you raised, as these will shape the direction of our future work.
>
> > it would be extremely helpful to also compare its performance to the vanilla VAE and other types of generative models, like diffusion models (which the authors mention)
>
> We agree a comparison with vanilla VAE would be helpful, but we do not think a comparison with diffusion models fits the intended scope of our paper, which is inference (and not generative performance). We realize this was unclear in our submission and will communicate our intention more clearly in future revisions.
>
> ### **Compositional code:**
>
> We did not clearly specify what we meant by composition code, and your criticism revealed that it was a crucial omission. The term "compositional code" can indeed mean different things in different contexts. In this work, we simply meant that the Gabor-like features shown in Figure 4 can be used to *compose*, or generate any black-and-white natural image. A model equipped with such a universal basis, and a good inference algorithm, should be able to perform accurate inference on any input and reconstruct it. This version of compositional is strongly supported by our results, throughout. We think the emergence of such a compositional code from MNIST is non-trivial and impressive. However, compositional could also mean learning abstract and modular representations in the latent space that transfer across tasks, which we did not test for. In the future, we will clarify what we mean by compositional, and possibly explore compositionally more generally, in the latent code sense and beyond.
>
> ### **Markovian assumption and the structure of the generative model:**
>
> Thank you for pointing this out. We recognize that our description of the Markovian assumption was unclear and could have led to confusion. In hindsight, we now recognize that it was incorrect to describe the marginal p(x) as Markovian, and this misrepresentation affected how the generative model was introduced. While we are still working through some ambiguities in our model's structure, we agree that presenting it more clearly would help. Specifically, we plan to include a graphical model in future versions to better illustrate both the generative and inference models. We appreciate your feedback in highlighting this area for improvement.
>
> ### **How is iP-VAE "brain-like":**
>
> A quantitative measure of "brain-like-ness" is arguably one of the most pressing open questions in our field. Therefore, we should not expect to solve this in one go. We agree that, given the importance of brain-like-ness, we should be clear about what we mean by it, and be quantitative whenever possible.
>
> In our work, we consider iP-VAE to be more brain-like than both vanilla (Gaussian) VAE and P-VAE. This is because iP-VAE uses integer spike count representations (as opposed to continuous, unbounded Gaussian latents), and it has an adaptive iterative inference algorithm (as opposed to the amortized inference utilized by the P-VAE). In the future, we plan to compare more thoroughly to alternative models, including the sampling-based approaches or more recent variants of predictive coding, to better situate our model within this vast literature in terms of the models’ brain-like-ness.

---

> ### Author Response · Authors · 2024-11-26
> **Clarifications and future directions in response to reviewer's feedback (2/2)**
>
> ### **What we meant by "prescriptive":**
>
> A lack of definition and motivation behind the word "prescriptive" is perhaps the most important missing piece from our Discussion section. We thank the reviewer for highlighting this and would like to clarify our intended meaning below.
>
> In our work, we started from a fairly general principle of ELBO maximization (or free energy minimization), added Poisson assumptions, and *derived* a set of dynamic equations that resembled a spiking neural network performing Bayesian inference through its membrane-potential-like update rule. In contrast, most work in the literature, including the paper mentioned by the reviewer, go in the reverse direction. Namely, they start with a set of dynamical equations and manipulate them to make their model perform Bayesian inference. In our view, only one of these directions is prescriptive: the one that goes from generic principles and derives dynamics from them, as opposed to putting dynamics in by hand.
>
> In physics, this is analogous to starting with the Principle of Least Action and selecting Lagrangian terms based on symmetry principles. This approach dictates the structure of the Lagrangian entirely from symmetry, determining not only which objects appear but also how they interact. Thus, fully deriving dynamics equations from first principles. This is such an elegant approach, and we wish to reproduce something like this in the messy world of NeuroAI. We do not expect this approach to translate cleanly to NeuroAI, nor do we believe that our current submission delivered a prescriptive approach comparable to those in physics. Nevertheless, we maintain that our work is more prescriptive than alternatives, and it paves the way for future developments to make it even more prescriptive. We will make sure to discuss this point in our future submission.
>
> ---
>
> Your insights are invaluable in refining our ideas, and we would be grateful for any further thoughts you might share on our responses. Thank you again for your time and constructive feedback. We greatly appreciate it.

---

> > ### Comment · Reviewer_hNnm · 2024-12-03
> >
> > I thank the authors for their effort and hope they are not too discouraged. There are some good ideas in the paper, but much more work is necessary. Some responses to comments:
> >
> > > The term "compositional code" can indeed mean different things in different contexts. ... However, compositional could also mean learning abstract and modular representations in the latent space that transfer across tasks, which we did not test for. ...
> >
> > I think this notion of "compositional" is not that interesting. By this definition, any performant autoencoder-type network exhibits a compositional code, since latent features (of some kind) are combined to reconstruct an input. At least to my knowledge, typical uses of "compositional" carry additional baggage and connotations, e.g., the code is modular in various senses. The code may indeed be modular somehow, and may generalize well across different data sets; my point is just that additional work needs to be done to show this. Section titles like "A compositional code that generalizes across domains" seem, given this issue, to suggest that the paper is making overly strong claims.
> >
> > > A quantitative measure of "brain-like-ness" is arguably one of the most pressing open questions in our field. Therefore, we should not expect to solve this in one go. We agree that, given the importance of brain-like-ness, we should be clear about what we mean by it, and be quantitative whenever possible. ...
> >
> > I understand the senses in which the proposed model is "brain-like". All I am arguing is that there are many such axes along which one can make these arguments, and it is not totally clear which, if any, are 'best'. Moreover, many of these models are incommensurate with respect to their "brain-like-ness" (e.g., maybe their networks solve different problems, or face different constraints, or operate at different scales).
> >
> > I agree that it is not reasonable to expect anyone to solve this problem in one go. But I still think claiming, e.g., that "These findings suggest ... provides a solid foundation for developing prescriptive theories in NeuroAI." sounds too much like claiming that the work in some sense presents an idea more "brain-like" than what has come before. Even if this is true, it requires a lot of supporting evidence to argue for. (Although I am skeptical that the notion of "most brain-like" without a lot of additional qualification even makes sense.)
> >
> > > A lack of definition and motivation behind the word "prescriptive" is perhaps the most important missing piece from our Discussion section. We thank the reviewer for highlighting this and would like to clarify our intended meaning below. ...
> >
> > Prescriptive theories (also called "normative" theories) are not alien to neuroscience, and well-known examples include efficient coding and Bayesian models of perception, both of which you are most likely familiar with. Efficient coding involves positing some kind of neural-response-dependent objective, and asserting that neurons respond in a way that maximizes it (given some set of constraints). See, e.g., Park and Pillow. Bayesian models of perception posit, e.g., that stimuli are combined or segregated according to the principles of Bayesian inference, and in particular that uncertainty strongly impacts this process. There's no objective function in that case (although you can probably formulate things in terms of one if you prefer) but the model still indicates how something 'ought' to be done.
> >
> > I hesitate to call the current work prescriptive for the following reason. It is true that there is an objective (ELBO) and assumptions/constraints (a certain structure involving Poisson latent variables). What makes prescriptive theories interesting, and adds force to the term 'prescriptive', is that you somehow 'get out' more than you 'put in'. Here, the form of updates follows fairly straightforwardly from ELBO + the Poisson-related assumptions. The final form of the proposed updates is not even quite what the model suggests it should be: to make things more biologically plausible, the authors simplify the update in certain ways. Without these simplifications, one doesn't get the 'nice' final answer. I am left feeling like most of the work was done by the (semi-arbitrary) choices made initially—and why ELBO, or Poisson, or whatever else? Do things change substantially if the latents are chosen to be discrete, but not Poisson? This is putting aside issues with the generative model, which you have acknowledged.
> >
> > When I think of "prescriptive" in the broader context of neuroAI, I also think questions like the following: what objective should I choose? What architecture should I choose? How should I model neurons? (Or if not neurons, what should I model?) Because this work is silent on such (extremely difficult) questions, I think it's not appropriate to present it as providing a prescriptive theory for the incredibly broad concept of "brain-like inference".

---

### Official Review · Reviewer_uBKc · 2024-11-09

**Soundness:** 3
**Presentation:** 3
**Contribution:** 2
**Rating:** 5
**Confidence:** 3

**Summary:**

The paper introduces Iterative Poisson VAEs (IP-VAEs) for performing approximate Bayesian inference. The work builds on recently proposed P-VAEs, which use Poisson distributions to approximate the posterior and thus employ discrete spike counts for inference. A major limitation of P-VAEs is their large amortization gap, which the authors address here through iterative inference. Unlike the original one-shot P-VAE, iP-VAE refines its representations over multiple iterations. Empirically, the authors demonstrate that this iterative approach enables an explicit trade-off between performance and computation while requiring significantly fewer parameters than competing methods. The model shows strong performance across several benchmark datasets, particularly closing the performance gap with sparse coding that existed in the original P-VAE.

**Strengths:**

- The paper adds to the literature of biologically-inspired inference. The extension introduced is simple and elegant and allows generalizing P-VAE to dynamically trade-off performance for compute.

- The experiments show that despite IP-VAE being order of magnitude more parameter efficient, they show significant improvement in terms of reconstruction error both for in- and out-of-distribution data.

-The general presentation is good and the paper is easy to follow.

**Weaknesses:**

- The biggest limitation is the large number of test-time iterations required to perform inference on relatively simple, downsampled datasets. While the authors mention plans to extend the approach to sequential data in future work, there is almost no discussion of the computational implications of these high iteration counts, which could potentially prohibit real-world applications.

- The authors only compare performance against P-VAEs and other variants of iterative VAEs. It would be valuable to include well-tuned baselines with standard VAEs (amortized, Gaussian distributions) to better contextualize the model's performance.

**Questions:**

- Why does the model require so many iterations (1000) at test time when trained with far fewer iterations? The significant disparity between training and test-time iterations needs theoretical justification.
- How does the performance scale with the dimensionality of the latent space? How would that affect the sparsity performance? It is difficult to assess this point without conducting a sensitivity analysis on the dimension of the latent states.
- In the appendix, the authors question the validity of the Poisson assumption, which is a known issue in computational/systems neuroscience. It would be valuable to move part of this discussion to the introduction to better contextualize the model's assumptions and limitations.
- How prone are the models to posterior collapse? Given that the objective penalizes large rates, how robust is the training of these models? Can the authors comment on the sensitivity of hyperparameters?

---

> ### Author Response · Authors · 2024-11-26
> **Our response to some of the concerns**
>
> Thank you for your thoughtful feedback. We greatly value your insights and have addressed some of the key concerns below.
>
> > The biggest limitation is the large number of test-time iterations required to perform inference on relatively simple, downsampled datasets
>
> We agree and thank the reviewer for raising this point. Upon reflection, we realized that our dynamics, in their current form, correspond to a massless particle with no momentum. To address this issue, we plan to experiment with the addition of higher-order terms yielding momentum.
>
> > It would be valuable to include well-tuned baselines with standard VAEs (amortized, Gaussian distributions) to better contextualize the model's performance.
>
> Good point. We will include Gaussian VAEs as a standard baseline.
>
> > How does the performance scale with the dimensionality of the latent space?
>
> Another interesting point that we are planning to explore in the future.
>
> > In the appendix, the authors question the validity of the Poisson assumption, which is a known issue in computational/systems neuroscience. It would be valuable to move part of this discussion to the introduction to better contextualize the model's assumptions and limitations.
>
> Agreed. We plan to include the essence of the argument in the introduction.
>
> > How prone are the models to posterior collapse?
>
> An interesting finding about P-VAEs is that they result in very few dead neurons. See Figures 4 and 8, and Table 2, in the original paper introducing P-VAE ([Vafaii et al., 2024](https://openreview.net/forum?id=ektPEcqGLb)). We plan to perform similar analyses on the iterative version to quantify this.

---

### Note · Authors · 2024-11-25

**Comment:**

We sincerely thank our reviewers, especially hNnm, for their time and insightful feedback. We believe that addressing these comments will greatly improve the quality of our paper.

After careful consideration of the reviews and the scope of revisions required, we have decided to withdraw this submission. We plan to resubmit a significantly enhanced version of the paper in the future.

We value the time and expertise invested in reviewing our submission and will ensure that the reviewers' insights are fully incorporated into our future work.

**Withdrawal Confirmation:**

I have read and agree with the venue's withdrawal policy on behalf of myself and my co-authors.

---

> ### Note · Program_Chairs · 2024-11-26
>
> We approve the reversion of withdrawn submission.

---

> > ### Author Response · Authors · 2024-11-26
> > **We are withdrawing this submission but hope to discuss with our reviewers**
> >
> > We decided to withdraw this submission but would still like to engage with the reviewers' valuable feedback and share our responses to their comments.
> >
> > Unfortunately, withdrawing prevents further discussion on openreview. To facilitate this, we requested the Program Chairs to reverse the withdrawal, and we're grateful they approved.
> >
> > **Dear reviewers:** Below, we'll share our prepared responses to your feedback. If you have time to reply, we'd greatly appreciate it, as your insights are invaluable for improving our work. Regardless, we deeply thank you for your time and thoughtful reviews.

---

### Meta-Review · Area_Chair_NRru · 2024-12-21

**Metareview:**

The paper was withdrawn by the authors.

**Additional Comments On Reviewer Discussion:**

see above

---

### Decision · Program_Chairs · 2025-01-22

Reject